# *Festuca coelestis* Increases Drought Tolerance and Nitrogen Use via Nutrient Supply–Demand Relationship on the Qinghai-Tibet Plateau

**DOI:** 10.3390/plants12091773

**Published:** 2023-04-26

**Authors:** Ningning Zhao, Xingrong Sun, Shuai Hou, Sujie Ma, Guohao Chen, Zelin Chen, Xiangtao Wang, Zhixin Zhang

**Affiliations:** 1College of Resources and Environment, Tibet Agriculture and Animal Husbandry University, Nyingchi 860000, China; 2Qiangtang Alpine Grassland Ecosystem Research Station (Jointly Built with Lanzhou University), Tibet Agricultural and Animal Husbandry University, Nyingchi 860000, China; 3College of Animal Science, Tibet Agricultural and Animal Husbandry University, Nyingchi 860000, China; 4College of Grassland Agriculture, Northwest A & F University, Yangling 712100, China

**Keywords:** *Festuca coelestis*, water–fertilizer coupling, nutrient utilization, physiological adaptation, root characteristics, agronomic efficiency of nitrogen

## Abstract

Drought and nutrient deficiency pose great challenges to the successful establishment of native plants on the Qinghai-Tibet Plateau. The dominant factors and strategies that affect the adaptation of alpine herbs to dry and nutrient-deficient environments remain unclear. Three water gradients were established using two-factor controlled experiments: low water (W_L_), medium water (W_M_), and high water (W_H_). The field water-holding capacities were 35%, 55%, and 75%, respectively. Nitrogen fertilizer (N) was applied at four levels: control (CK), low (F_L_), medium (F_M_), and high (F_H_) at 0, 110, 330, and 540 mg/kg, respectively. The results revealed that N was the main limiting factor, rather than phosphorous (P), in *Festuca coelestis* under drought stress. Under water shortage conditions, *F. coelestis* accumulated more proline and non-structural carbohydrates, especially in the aboveground parts of the leaves and stems; however, the root diameter and aboveground nitrogen use efficiency were reduced. Appropriate N addition could mitigate the adverse effects by increasing the release of N, P, and enzyme activity in the bulk soil and rhizosphere to balance their ratio, and was mainly transferred to the aboveground parts, which optimized the supply uptake relationship. The effects of water and fertilizer on the physiological adaptability and nutrient utilization of *F. coelestis* were verified using structural equation modeling. Based on their different sensitivities to water and nitrogen, the W_H_F_M_ treatment was more suitable for *F. coelestis* establishment. Our results demonstrated that the disproportionate nutrient supply ability and preferential supply aboveground compared to below ground were the main factors influencing *F. coelestis* seedling establishment under drought conditions. This study provides evidence for a better understanding of herbaceous plants living in high mountain regions and offers important information for reducing the risk of ecological restoration failure in similar alpine regions.

## 1. Introduction

Reseeding with native plant species has been widely applied for restoring degraded grasslands because native plants are well adapted to local soil and climatic conditions and are thus easily established [1]. However, large seasonal variations in climatic conditions often lead to low survival rates and poor establishment, and the maintenance of water and nutrients is a prerequisite. Water carries soil nutrients and is critical for nutrient transfer and uptake. Increasing the soil moisture content can promote the mineralization of organic nitrogen [2], and there is a close relationship between the soil nitrogen availability and soil moisture content [3]. Low soil moisture conditions influence nutrient availability and their transfer rate to the root system. Fertilization can promote grass growth, improve photosynthesis, and increase the efficiency of water and fertilizers, thereby enhancing the accumulation of photosynthetic compounds [4]. The interaction between water and nitrogen can affect the nitrogen use efficiency. Therefore, water–fertilizer coupling is one of the most important factors for plant establishment, growth, and development [2].

Changes in water and nutrient conditions affect photosynthetic processes during plant growth. Non–structural carbohydrates (NSCs) and proline play critical roles in plant physiological processes. Non-structural carbohydrates are crucial for fulfilling multiple functions in plants, including metabolism, transport, osmoregulation, and export as substrates for soil organisms. They are primarily composed of soluble sugars and starch. Starch supplies plants with energy and carbon resources. The released sugars and other derived metabolites support plant growth and mitigate the negative effects of stress by acting as osmotic protective substances and compatible solutes [5]. Proline and soluble sugars, which are important osmosis-regulating substances in plants, enable plants to adapt to changes in external environmental conditions by reducing their osmotic potential [6]. Drought–affected plants accumulate large amounts of proline and soluble sugars to adapt to drought, which affects their photosynthetic capacity and starch synthesis. According to previous studies, the total starch and maximum and average accumulation rates decrease in drought-affected sorghum during the middle and late filling stages [7]. Under various irrigation levels, plants fed with an appropriate amount of nitrogenous fertilizer accumulate more proline and soluble sugars in the leaves, whereas overfertilization inhibits the increase in soluble sugar content [8]. However, the expression and transfer of soluble sugars and proline in different plant components require further elucidation.

The root is the main organ for water and nutrient absorption [9], and its growth and morphological characteristics are affected by environmental changes. Nutrient absorption is severely hindered when soil moisture deficit intensifies. Under such conditions, plants reduce or even stop the growth of the root system to maintain the water required for their metabolism, thereby affecting normal plant growth [10,11]. As the soil moisture content increases, the crop roots and aboveground parts grow faster, and the demand for water and fertilizer increases. At this time, the root biomass increases rapidly, and the water and nutrients absorbed by some roots can support the growth of the entire plant. However, the root systems of plants in low-fertility soils must be further developed to ensure the availability of the nutrients required for plant growth under high moisture conditions [12]. Therefore, changes in the growth and morphology of the root system reflect the adaptive strategies of plants for water and nutrient uptake.

Soil nutrient availability is another key factor affecting plant adaptability. The supply of nutrients to the soil depends on the activity of soil enzymes and changes in the rhizosphere. Soil enzymes link plants to available soil nutrients. Urease plays a critical role in soil N cycling by releasing inorganic N for plant uptake [13]. Phosphatases play a major role in regulating phosphorus nutrition, metabolism, and organic phosphorus reuse. It can convert organic phosphorus in plants into inorganic phosphorus, which is transported from aging to young tissues. Moreover, their activities directly affect the availability of organophosphates [14]. In the desert grass of Inner Mongolia, nitrogen application improved the soil enzyme activity under drought [15]. Soil enzymes are more active in the rhizosphere and can improve the efficiency of soil nutrient activation and release. In the rhizosphere environment, fine roots transfer carbon, nitrogen, and some molecular substances into the soil through rhizosphere deposition (root exudates, lysates, mucus, and exfoliation of cells and cortical tissues), altering soil pH, nutrients, and enzyme activities, and leading to changes in the physical, chemical, and biological properties of the rhizosphere and the original soil mass [16]. Different water and fertilizer environments may change the quantity and composition of plant root exudates, the activity pattern and function of rhizosphere soil enzymes, and the cycle and availability of soil nutrients, thereby affecting plant growth [17]. Nevertheless, the regulatory mechanisms of herbaceous plants on plateaus in response to nitrogen, phosphorus, and other water–fertilizer conditions remain unknown.

The alpine meadow steppe, one of the main grassland types in the Qinghai-Tibet Plateau, is vulnerable to disturbance because of its fragile ecological environment and the weak self–healing and regulatory abilities of the native vegetation community [18]. According to previous studies, the ecological restoration effect was significantly reduced in the second year when *Slender wildrye*, *Elymus dahuricus* Turcz, *Bromus inermis* Leyss, and *Poa crymophila* were selected for reseeding in the northern Tibetan Steppe [19]. *Festuca coelestis*, the dominant native plant in alpine meadow steppes, has good adaptability to the environment. It has been applied to ecological restoration and has achieved favorable results. However, poor growth potential at the seeding stage and low survival rates persist during the early stages of ecological restoration.

Previous studies have revealed the physiological traits of adaptation to native grass in alpine regions; however, the supply of nutrients from the soil has been less considered. The species used in the ecological restoration of Tibet have focused on the influence of a single factor: water–nitrogen coupling. The nutrient release and absorption processes under the interaction of water and nitrogen are unclear. Therefore, based on the relationship between water and nutrients, this study aimed to improve the role of native plants in ecological restoration. A greenhouse trial was conducted under different water and fertilizer conditions to investigate (1) NSCs and proline in the early stages of establishment under the interaction of water and nitrogen and the supply relationship of N and P, (2) the effect of water and fertilizer treatments on the root growth of native plant species, and (3) the rhizosphere and non-rhizosphere nutrient supply modes. The results of this study are expected to clarify the physiological response of typical plants in alpine meadows to the interaction of water and nitrogen and provide a reference for the rational use of native plant species for ecological restoration.

## 2. Results

### 2.1. Effect of Water and Fertilizer Treatments on the Physiological Regulation of Festuca coelestis

The NSC contents in the different biological components of *F. coelestis* under different water and fertilizer treatments varied significantly (*p* < 0.05). Overall, the contents of non-structural carbohydrates, soluble sugars, and starch in *F. coelestis* under the CK were the highest and increased with intensified water stress but decreased with an increase in the fertility gradient. The contents of non-structural carbohydrates and soluble sugars in *F. coelestis* under W_L_CK reached the highest point, whereas the starch content under W_L_CK was the highest.

The contents of NSCs, soluble sugars, and starch in the leaves and stems of *F. coelestis* were significantly affected by water and fertilizer application, whereas those in the roots remained stable. The contents of NSCs in the leaves decreased significantly with increasing fertility (*p* < 0.05), and the range of decrease was 5.65–148.8% (*p* < 0.05). The NSC contents of stems under different water treatments varied significantly. The contents of NSCs in stems under F_M_ were the lowest under medium and low soil moisture conditions (W_M_ and W_L_) and lower by 48.5% and 41.2%, respectively, than that of CK (*p* < 0.05). The soluble sugar content in the leaves under the F_M_ was 187.1% lower than that under the W_M_ (*p* < 0.05). Under W_L_, the soluble sugar content in the leaves and stems decreased significantly with an increase in fertility, being 43.2% and 89.0% lower, respectively, than that of the CK (*p* < 0.05), in the case of F_H_. The starch content in the stems decreased with increasing fertility. More precisely, the starch content in the stems under W_M_, W_L,_ and in leaves under F_H_ was lower by 58.5%, 61.9%, and 85.9%, respectively, than that under CK (*p* < 0.05).

The proline content of the different biological components of *F. coelestis* under different water and fertilizer treatments varied significantly (*p* < 0.05). Overall, the proline content in *F. coelestis* under F_H_ was the highest, with an increasing trend over the water stress and fertility gradients. The content of proline in *F. coelestis* under W_L_F_H_ reached its highest point, which was 39.6% higher than that in CK (*p* < 0.05). The proline content in the leaves and stems was significantly affected by water and fertilizer, whereas that in the roots remained stable. The proline content in leaves and stems increased significantly with increasing fertility (*p* < 0.05). The proline content in the leaves of *F. coelestis* under W_M_ and W_L_ conditions in the case of F_H_ was higher by 13.5% and 35.1%, respectively, than that in CK (*p* < 0.05). The proline content in the stems of *F. coelestis* under W_H_ and W_L_ conditions in F_H_ was higher by 57.5% and 29.7%, respectively, than that in CK (*p* < 0.05) (Figure 1).

### 2.2. Effect of Water and Fertilizer Treatments on Root Characteristics of Festuca coelestis

The root traits of *F. coelestis* changed with water and fertilizer conditions (*p* < 0.05) and were clear for both the root length and surface area. Root growth was influenced more by water and declined with water stress. The root surface area, total root length, and root volume under F_H_ were higher by 174.2%, 303.0%, and 466.7%, respectively, than those under W_L_ (*p* < 0.05). The root surface area, total root length, and root volume under W_M_F_H_ reached the maximum, being 213.5%, 142.8%, and 255.4% higher, respectively, than those under the CK (*p* < 0.05). The root surface area and total root length in the W_H_ decreased and then increased as the fertility gradient improved. The root surface area and total root length under F_L_ were higher by 258.6% and 171.9%, respectively, than those under CK (*p* < 0.05). In contrast, the average root diameters in W_L_ and W_H_ first increased and then decreased as the fertility gradient increased. The average root diameter under W_H_F_M_ was 52.2% higher than that under the CK (*p* < 0.05). The root volumes of *F. coelestis* in W_H_ and W_M_ increased with increasing fertility. The average root diameter and root volume in the W_L_ and W_m_ treatments were less affected by fertility (Figure 2).

### 2.3. Effects of Water and Fertilizer Treatment on Soil N and P

The effects of water and fertilizer on the available nitrogen and urease activities varied, and the rhizosphere and bulk soils indicated significant differences (*p* < 0.05). Regulated by water and fertilizer treatments, the available nitrogen content in the rhizosphere and bulk soils decreased with decreasing soil moisture content, and the urease activity was less affected by water. Sensitivity under different fertility conditions varied, first increasing then decreasing or first decreasing and then increasing overall. Specifically, the available nitrogen content in the rhizosphere and bulk soils initially increased and then decreased with increasing fertility. The available nitrogen content in the bulk soil under W_L_ alone was the highest when the fertilizer level was high (F_H_); in other cases, the content reached the highest point when the fertilizer level was medium (F_M_). The urease activity in the rhizosphere and bulk soils under W_H_ were the highest when the fertilizer level was high (F_H_) and were 9.3% and 2.7% higher, respectively, than that of CK. It was less affected by fertility under the W_L_. The available nitrogen content and urease activity in the bulk soil under W_H_F_M_ and rhizosphere soil under W_M_F_L_ reached their highest points, being 90.6% and 26.7% higher, respectively, than those of CK (*p* < 0.05) (Figure 3).

The effects of water and fertilizer treatments on the content of rapid available phosphorus and alkaline phosphatase in *F. coelestis* varied, and the rhizosphere and bulk soils showed significant differences (*p* < 0.05). The alkaline phosphatase content decreased with decreasing water content. The sensitivity varied under different fertility conditions. In the case of rapidly available phosphorus, it first increased and then decreased or first decreased and then increased overall. The alkaline phosphatase levels gradually decreased, initially increased, and then decreased.

Specifically, the rapidly available phosphorus content in the rhizosphere and bulk soils under W_H_ initially increased and then decreased with increasing fertility. The contents of the rapidly available phosphorus in the rhizosphere and bulk soils under F_L_ were 82.8% and 6.7% higher, respectively, than that under CK (*p* < 0.05). The alkaline phosphatase content in the rhizosphere soil under W_H_ increased with increasing fertility. The content of alkaline phosphatase in the rhizosphere soil under F_H_ was 82.5% higher than that under CK (*p* < 0.05). Under (W_L_), the content of rapidly available phosphorus in the rhizosphere soil under F_M_ and bulk soil under F_L_ reached its highest point, which was 28.5% and 169.1% higher than that of the CK, respectively (*p* < 0.05). In the W_M_, the rapidly available phosphorus decreased as fertility increased (*p* < 0.05). The content of alkaline phosphatase in the rhizosphere soil under F_M_ reached its highest point under medium and low soil moisture conditions (W_M_ and W_L_), and was 75.2% and 122.3% higher than that of CK, respectively (*p* < 0.05). The contents of rapidly available phosphorus and alkaline phosphatase in the bulk soil under the W_M_CK and W_H_F_L_ treatments reached their maximum values (Figure 4).

### 2.4. Effect of Water and Fertilizer Treatments on Agronomic Efficiency of Nitrogen of Festuca coelestis

The agronomic efficiency of nitrogen refers to the ratio of nitrogen absorbed by fertilized crops to the amount of nitrogen applied. AEN = (crop yield in N application zone—crop yield in N free zone)/N fertilizer input. The agronomic efficiencies of nitrogen in *F. coelestis* under different water and fertilizer treatments varied (*p* < 0.05). In general, the aboveground agronomic efficiency of nitrogen decreases with an increase in water stress and fertility. The agronomic efficiency of nitrogen in *F. coelestis* under F_H_ was 6–18 times lower than that of *F. coelestis* under F_L_ (*p* < 0.05). The underground agronomic efficiencies of nitrogen of *F. coelestis* under F_L_ and F_M_ decreased and then increased with intensified water stress, whereas those of *F. coelestis* under W_H_ and W_M_ increased with increasing fertility (*p* < 0.05). The aboveground and underground N agronomic efficiencies reached their maxima under the W_H_F_L_ and W_L_F_L_, respectively. Nevertheless, the underground agronomic efficiencies of nitrogen under W_H_F_L_ and W_M_ were negative under F_L_ and F_M_ (Figure 5).

### 2.5. Effects of Water and Fertilizer Treatment on Nutrient Cycling and Physiological Characteristics of Festuca coelestis

In this model, water (standardized effect = −0.828) exerted a significant negative effect on the size of physiological characteristics, but physiological traits had no significant effect on the nitrogen use efficiency (*p* > 0.05). Nitrogen (−0.470) negatively affected the available phosphorus, and the available phosphorus (−0.232) negatively affected nitrogen use efficiency. Water (0.533) and nitrogen (0.631) exerted significant direct positive effects on the available nitrogen, whereas the available nitrogen (0.821) had a significant positive effect on the nitrogen use efficiency. Nitrogen (0.244) exerted a positive effect on the N use efficiency, whereas water had no significant effect (*p* > 0.05). However, water had an indirect effect on the N use efficiency through available nitrogen (0.437) (Figure 6).

## 3. Discussion

In the extreme environment of drought and nutrient depletion of the soil in the alpine region, drought inhibits important metabolic processes in plants and biochemical reactions in the soil, and nutrient deficit limits the water utilization efficiency, thereby affecting plant growth and development. By considering plant and soil properties, this study elucidated the changes in the physiological adaptation and nutrient utilization of *F. coelestis* under different water and fertilizer treatments.

Water and fertilizers can affect the physiological activities of plants. When plants are under drought stress, their self-protection system activates the osmotic regulation system. Consequently, various solutes accumulate in the cells to reduce the cell osmotic potential and increase osmotic pressure to maintain the water-absorbing capacity and prevent dehydration. Proline, soluble sugars, and starch accumulate in plants under aggravated drought stress [20]. A similar pattern was observed in this study, which is consistent with the conclusion that water (−0.828) has a significant negative effect on the size of the physiological characteristics in the SEM. This pattern was the most significant in the leaves and stems of *F. coelestis*.

In the SEM, nitrogen had no significant effect on physiological characteristics (*p* > 0.05), which may be because of the different effects of nitrogen fertilizer on proline, starch, and soluble sugars. Specifically, in the leaves and stems of *F. coelestis*, the proline content increased as the fertility gradient increased under different water treatments, similar to the findings of Kumari et al. (2015). The application of nitrogen fertilizer promotes the accumulation of proline. Nitrogen fertilizer can enhance the permeability of cells, increase the concentration of cell fluid, decrease the water potential, enhance the water retention capacity, maintain a balance with the environment, and offset the damage caused by stress to plants [21]. The soluble sugar and starch contents changed slightly as the fertilizer gradient increased in W_H_. Thus, when water is abundant, high nitrogen levels do not improve the transport of photosynthetic products [22]. The soluble sugar and starch contents under W_L_ decreased as the fertilizer gradient increased, similar to the results of Yu et al. (2018). Adding exogenous nitrogen likely promotes photosynthesis; plants grow rapidly, and more energy is transferred to the roots for the absorption and transport of mineral nutrients when the leaves and stems meet their own needs [23]. However, the soluble sugar content in leaves and stems is much higher than that in roots, which may be because in the early stage of plant establishment, root growth and maintenance of respiration consume a high amount of energy [24]. The starch content in stems was much higher than that in leaves and roots, likely because stems are responsible for storage and transportation and are closest to the leaves. According to the principle of nearby distribution of carbohydrates from carbon sources to carbon sinks, stems are the first to obtain starch for growth, development, and storage [25].

The root is an organ in which plants absorb and transport water and nutrients. Plant growth and morphology are strongly affected by water and fertilizer conditions. Changes in root morphology can affect the absorption rate and probability of water and nutrient acquisition. A drought environment limits the growth and development of the root system, whereas the addition of nitrogen can induce root system growth and development [26]. The present study indicated that the root surface area, total root length, and root volume of *F. coelestis* decreased as the water stress intensified, which is similar to the findings of Bengough et al. The root surface area, total root length, and root volume of *F. coelestis* under W_L_ were reduced by F_H_, similar to the findings of Wang et al. Under severe water deficit, the soil water potential is low, and the root system in dry soil is usually limited by mechanical resistance and water stress, thus hindering the growth and development of the root system [27]. In addition, excess nitrogen in a low volume of water can hardly be fully dissolved, making it more difficult for plants to absorb N. Consequently, the activity of enzymes related to plant respiration is inhibited, affecting nitrogen metabolism. Excess unused nitrogen results in lower soil pH, soil acidification, and nitrogen agronomic efficiency, thereby inhibiting plant root growth [28,29]. In addition, water can accelerate the dissolution of fertilizer so that nutrients can be fully released and migrated, and nitrogen fertilizer can increase the content of soil nutrients, improve the availability of soil nutrients and water utilization efficiency, promote root respiration, and promote root system development [30].

Water is a carrier of nutrients, whereas fertilizers provide nutrients to plants; both are indispensable for plant growth and development [31]. The rhizosphere environment of vegetation changes owing to the supply of water and fertilizer to the soil, which affects material exchange and energy transfer between plants and soil. In addition, the soil enzyme activity was the highest in the rhizosphere. Adequate fertilization can significantly improve the activity of rhizosphere enzymes, thereby promoting nutrient cycling and availability [32].

This study indicated that both water (0.533) and nitrogen (0.631) had significant direct positive effects on available nitrogen in the structural equation models. Specifically, the available nitrogen, urease, and alkaline phosphatase content decreased as the soil moisture content decreased, which is similar to the results of Zhu et al. (2008). Water may promote the growth of root systems, resulting in increased root exudates, improved rhizosphere enzyme activity, and higher agronomic nitrogen [15]. The fertilization of *F. coelestis* under different water treatments improved the available nitrogen content. However, the available nitrogen content decreased as the amount of applied fertilizer increased, possibly because urease activity was inhibited by the high nitrogen levels [33]. High nitrogen levels inhibited the rapid availability of phosphorus in the rhizosphere soil under W_H_ and W_L_, similar to the findings of Ladanai et al. A possible reason for this is that the addition of nitrogen improves the nitrogen-to-phosphorus ratio. As a result, more phosphorus is absorbed, resulting in a lower content of rapidly available phosphorus [34]. This can be verified by SEM results, which showed that nitrogen (−0.470) has a notable negative effect on available phosphorus, but not on water. Differences in available nitrogen and phosphorus content reflect the impact of different water and fertilizer conditions on the activation and release of soil nutrients [2]. Considering that the content of available nitrogen in the rhizosphere soil was higher than that in the bulk soil under W_L_F_M_ alone, appropriate fertilization created the best effect under W_H_ and W_L_ including rapidly available phosphorus. The available nitrogen and rapidly available phosphorus contents in the bulk soil under the W_H_F_M_ and W_M_CK treatments were the highest.

Fertilization improved the activity of alkaline phosphatase under the water treatment, whereas urease activity in the rhizosphere soil conformed to this rule under W_H_ alone. The urease activity in the bulk soil reached its highest level as the amount of fertilizer and water increased. This result is similar to that reported by Sun et al. A probable reason for this is that fertilization accelerates microbial reproduction and increases soil enzymes [35]. Plants require abundant water for N absorption. When water is sufficient, a larger amount of nitrogen fertilizer is applied, which means that more nitrogen is provided to the plant and more nitrate accumulates, thereby enhancing the activity of the enzymes involved in nitrogen metabolism [36]. However, the alkaline phosphatase activity in the bulk soil decreased as the amount of fertilizer increased, which is similar to the results of Keeler et al. Bulk soil is less affected by the root system; thus, soil acidification owing to higher nitrogen levels results in decreased alkaline phosphatase activity [37]. Urease activity in the rhizosphere and bulk soil under W_H_ reached its highest levels under F_H_. Appropriate fertilization led to the best effects under W_M_ and W_L_. Under different water treatments, a larger amount of fertilizer increased the alkaline phosphatase activity; thus, appropriate fertilization would have the best effect on bulk soil. The activities of urease and alkaline phosphatase were the highest in bulk soil under W_H_F_L_ and in rhizosphere soil under W_M_F_L_, respectively. This indicates that the activity of soil enzymes was affected by different water and fertilizer treatments. This may be because soil nutrients are released differently by water and fertilizer and affect the growth of plants and the production of root exudates, causing varied activities of soil enzymes [38].

The agronomic efficiency of nitrogen refers to the efficiency of plants in recovering nitrogen fertilizer applied to the soil [39]. The soil moisture content directly affects the availability of soil nitrogen and is the determining factor for the dissolution and mineralization of nitrogen fertilizers. Applying an appropriate amount of nitrogen fertilizer can increase the soil nutrient content and meet the needs of vegetation growth and development [4]. This conclusion was verified via SEM, which revealed that available N (0.821) and N (0.244) had positive effects on NUE, whereas water had an indirect effect on NUE through available N (0.437). Specifically, the aboveground agronomic efficiency of nitrogen in *F. coelestis* decreased with an increase in water stress and fertility, similar to the results of Rietra et al. (2017) and consistent with the change law of aboveground biomass [40]. This may be because the addition of appropriate nitrogen can increase the soil nitrogen content, whereas sufficient water can promote the dissolution and transport of available nitrogen, improve urease activity, provide sufficient nitrogen to plants, and promote plant growth. The agronomic efficiency of nitrogen increases [41]. Overfertilization affects plant growth, and the nitrogen applied to the soil cannot be completely absorbed. In addition, unused nitrogen fertilizer is lost through denitrification, ammonia volatilization, and leaching, ultimately leading to a significant reduction in the aboveground agronomic efficiency of nitrogen [42]. The underground agronomic efficiency of nitrogen in *F. coelestis* under W_H_ and W_M_ increased with increasing fertility. The underground agronomic efficiencies of nitrogen in *F. coelestis* under W_H_F_L_, W_M_F_L_, and W_M_F_M_ were negative, similar to the findings of Brown et al. (2019) and consistent with the change law of underground biomass [40]. When there is sufficient water, along with an increase in nitrogen fertilizer, the dissolution and transport rates of soil nutrients are high; therefore, the available nitrogen content, urease activity of urease, root respiration, and nutrient absorption are adequate. In this case, the root system grows well, the underground biomass continues to increase, and the agronomic efficiency of nitrogen improves [30]. When water is deficient, the ability of the root system to actively adapt to the environment decreases, resulting in reduced total root length, surface area, and volume, thereby inhibiting root growth and development [43]. Increasing the nitrogen level does not work in this case, as the fertilizer cannot be fully dissolved, and excessive concentrations make it more difficult for plants to absorb and inhibit the activity of related enzymes, thereby affecting nitrogen metabolism and inhibiting root growth [29]. Therefore, favorable water and fertilizer conditions can improve the agronomic efficiency of nitrogen.

Our results were based on water and nitrogen control experiments. During ecological restoration in the field, the interactions between different nutrients, especially nitrogen, phosphorus, and K, are extremely important. At present, we have studied the change law of nitrogen under drought; however, the mutual regulatory mechanisms of nitrogen, phosphorus, and potassium require further study. The process by which mycorrhizal fungi and microorganisms activate soil enzyme activity and increase soil organic matter content, thereby regulating the rhizosphere soil environment, supporting host plant access to nutrients and water, and alleviating various environmental stressors, is also noteworthy. In addition, as one of the important factors in plant establishment, growth, and development, the competition between reseeding plants and weeds deserves attention.

## 4. Materials and Methods

### 4.1. Experimental Site

The experimental site was located at the Pratacultural Science Practice Base of the Tibet Agricultural and Animal Husbandry University, Bayi Town, Bayi District, Linzhi, Tibetan Autonomous Region, China. The study area has a temperate plateau humid/semi-humid monsoon climate, and the peak rainy season generally occurs between the end of June and the end of August. The mean annual temperature in the study area is 7~16 °C, the mean annual precipitation is 650 mm, the total annual solar radiation is 5460~7530 MJ/m^2^, and the mean annual relative humidity is 71%. The soil nutrients in this area were poor and the loss of nitrogen and phosphorus was significant. The test material was *F. coelestis*, which is a staple gramineous pasture found in the alpine grasslands of northern Tibet. It was collected in Nagqu, Tibet, in September 2020 and stored at 5 °C before the experiment.

### 4.2. Experimental Design and Sample Collection 

#### 4.2.1. Experimental Design

Pot experiments were performed in a greenhouse at 26 °C from August to December 2021. Cylindrical flowerpots with a height of 20 cm and an inner diameter of 28.5 cm were used. The soil for the test was sandy loam, and the soil samples were naturally air-dried after collection; the pot soil weighed 8.5 kg. The pots were filled with sandy loam (USA soil taxonomy) with a pH of 7.30, 25.73 mg/kg of fast-release nitrogen fertilizer, and 9.30 mg/kg fast–release phosphorus fertilizer. The field capacity (FC) and bulk density of the soil were 30.27% and 1.32 g/cm^3^, respectively. A two-factor, split–plot design was used. Three levels of soil moisture content were used: sufficient, mild, and moderate. The soil moisture content was set to 75%, 55%, and 35% of the maximum field water-holding capacity. N was applied at four levels: control (no fertilization), low, medium, and high, in which 0, 0.11, 0.33, and 0.54 g/kg, respectively, of pure nitrogen were utilized. A total of 12 treatments, sextuplets, and 72 pots were used.

Seeds were sown on 6 August 2022. After germination, five seedlings with consistent growth were placed in each pot, transplanted, treated with base fertilizer (phosphorus pentoxide: 0.07 g/kg and potassium chloride: 0.1 g/kg), and irrigated to maintain a 60% soil moisture content during this period. One week later, additional urea was applied according to the amount calculated based on the unit area for the different treatments, and water was supplemented according to the amount calculated based on the designed soil moisture content using the weighing method. The three plants were repositioned weekly to ensure consistency during the experiments.

#### 4.2.2. Sample Collection

Soil collection: the rhizosphere and bulk soils were collected after the experiment. The root-shaking method was adopted for rhizosphere soil. Three plants with uniform growth were randomly selected for each treatment, and the root system was removed from the soil. The loose soil attached to the root system was shaken off gently, and the bulk soil was collected at the same time and then quickly collected into separate plastic bags. When the soil samples were taken to the laboratory, a sterilized brush was used to remove the soil attached to the roots, and visible roots in the soil were removed. The collected soil was sieved (<1 mm) and placed in a refrigerator at 4 °C for the determination of soil nutrients and enzyme activity.

Collection of roots, stems, and leaves: three plants with uniform growth were randomly selected for each treatment. The roots, stems, and leaves of the processed plants were separated, wrapped with tin foil by valve bags to keep their shape intact, and then placed under −80 °C for determination of the plant enzyme activity and nutrient contents.

### 4.3. Measurement Items and Methods

#### 4.3.1. Determination of Soil Nutrients and Enzyme Activity

The collected soil was sieved (<1 mm) and placed in a refrigerator at 4 °C for the determination of soil nutrients and enzyme activity. The molybdenum-antimony resistance colorimetric method was used to determine the available phosphorus content, and the alkaline hydrolysis diffusion method was used to determine the available nitrogen content [44].

Urease activity was measured using a colorimetric method. A 5 g soil sample was incubated with 1 mL of methylbenzene, 10 mL of 10% urea solution, and 20 mL of pH 6.7 citrate buffer at 37 °C in an incubator. After incubation for 24 h, 4 mL of phenol sodium solution and 1 mL of hypochlorite solution were added to 3 mL of the suspension and then measured using a spectrophotometer at a wavelength of 578 nm within 1 h, and the alkaline phosphatase activity was measured using the disodium phenyl phosphate method. A 5 g soil sample was incubated with 20 mL of 0.5% disodium phenyl phosphate solution at 37 °C in an incubator. After incubation for 24 h, 2 mL of the ammonium molybdate solution was added to 10 mL of the suspension and measured using a spectrophotometer at a wavelength of 578 nm [45].

#### 4.3.2. Determination of Soluble Sugar, Starch, and Proline

In this analysis, the NSC represents the sum of soluble sugars and starch. The total soluble sugar and starch contents were measured using the anthrone–sulfuric acid method. Briefly, oven–dried samples (0.1 g) with 5 mL of 80% *v*/*v* ethanol were incubated at 80 °C for 40 min and then centrifuged at 8000× *g* for 10 min. The supernatant was collected, and the residue was re-extracted twice, as described above. The residue was retained for starch analysis, as described in this section. We added 5 mL of the anthrone reagent to the collected supernatant, incubated at 100 °C for 10 min, and then determined the absorbance at 620 nm using an ultraviolet–visible spectrophotometer. The proline levels were measured using acid ninhydrin colorimetry. A fresh 0.25 g sample was crushed in 10 mL of 3% aqueous sulfosalicylic acid and centrifuged at 1500× *g* for 10 min. Then, 2 mL of the supernatant was added to 2 mL glacial acetic acid, after which 2 mL acidic ninhydrin was added and kept at 100 °C in a bain-marie for a period of 60 min. The reaction was stopped by placing the mixture in an ice bath, and 4 mL toluene was added. The absorbance of the upper phase was measured at 520 nm using toluene as the blank [46].

#### 4.3.3. Determination of Root Indices

The root image was obtained using the root scanner, and then analyzed using Winrhizo Tron 2011 to obtain the root length, root surface area, root volume, and other morphological parameters.

### 4.4. Statistics and Data Analysis

The measured data were processed by Microsoft Excel 2010 and SPSS 21.0 (Version 21.0, IBM Corp., Armonk, NY, USA). One-way and two-way ANOVA analyses were conducted at the significance level of α = 0.05. The least significant difference (LSD) was adopted for multiple comparison of means. All analysis results were plotted using Origin2021 (Origin Lab Corporation, Northampton, MA, USA). To reveal the effects of water and fertilizer on nutrient cycling and physiological characteristics, an SEM analysis was undertaken with Amos software (17.0.2, IBM SPSS Inc., Armonk, NY, USA).

To reveal the effects of water and fertilizer on nutrient cycling and physiological characteristics, structural equation modeling analysis was performed using the Amos software (17.0.2, IBM SPSS Inc., Armonk, NY, USA), which can explain the regression relationship between observed variables to intuitively represent the response mechanism between variables through the path graph. We standardized all explanatory variables before the analysis. To divide the factors with the same category into a group (containing two or more factors), PCA was conducted with the packages “FactoMineR”, “factoextra”, and “corrplot” in R, and PCA was used to transform multiple factor variables into a set of variables (Appendix A).

## 5. Conclusions

The interaction between water and fertilizer plays an important role in regulating the growth of the dominant species *F. coelestis* in alpine grasslands. Under drought stress, the drought resistance of *F. coelestis* was improved by increasing the proline and nonstructural carbohydrates in the leaves and stems. Nitrogen addition was beneficial for the accumulation of proline and decreased the content of nonstructural carbohydrates in leaves and stems. The responses of roots to different water and nitrogen conditions were lower. Drought severely limits nutrient cycling, and F_M_ can produce a compensation effect. Specifically, with the intensification of drought stress decreasing available nitrogen and urease activity, the root growth was inhibited, resulting in lower NUE. However, nitrogen addition (F_M_) improved the nutrient release by increasing the content of available nitrogen and phosphorus in both the rhizosphere and bulk soil, improving the activities of urease and alkaline phosphatase, root growth, and NUE. In conclusion, this study demonstrates that water and nitrogen regulate the nutrient cycle of *F. coelestis* by enhancing physiological adaptability, promoting root growth, and balancing the nitrogen and phosphorus nutrient supply in the soil. The nutrient absorption and utilization effect of *F. coelestis* treated with W_H_F_M_ was the most significant, making it suitable for supplementary sowing in alpine meadows, providing a reference for the development and utilization of seed resources and the restoration of degraded grasslands.

## Figures and Tables

**Figure 1 plants-12-01773-f001:**
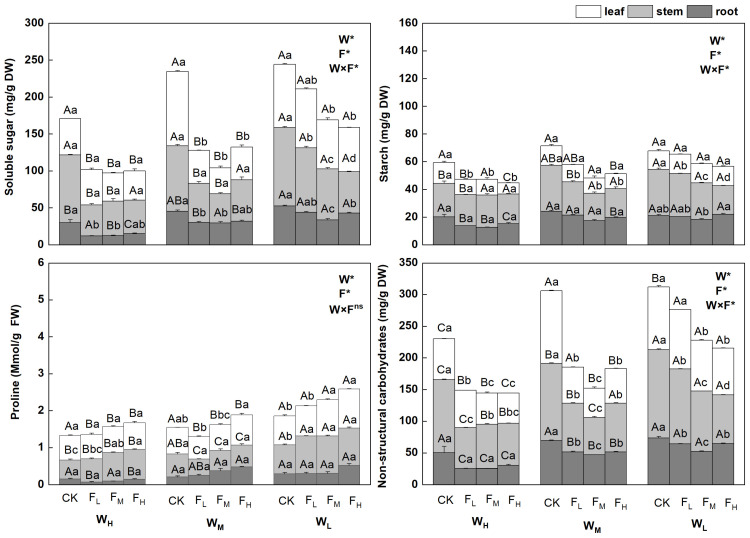
Changes in physiological characteristics of *Festuca coelestis* under different water and fertilizer treatments. Notes: W, F, and W×F indicate the effects of water and fertilizer and their interaction results from ANOVA (*p* < 0.05). Asterisks (*) and ns indicate significant differences. Uppercase and lowercase letters represent the differences between the water and fertilizer treatments.

**Figure 2 plants-12-01773-f002:**
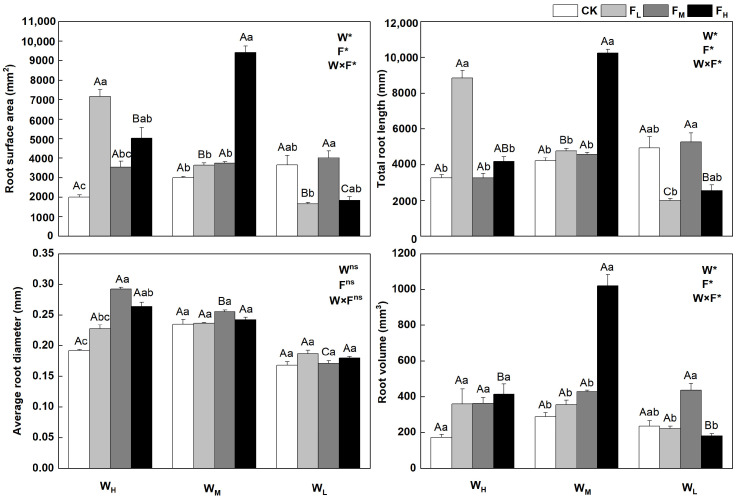
Changes in root system of *Festuca coelestis* under different water and fertilizer treatments. Notes: W, F, and W×F indicate the effects of water and fertilizer and their interaction results in the ANOVA (*p* < 0.05). Asterisks (*) and ns indicate significant differences. Uppercase and lowercase letters represent the differences between the water and fertilizer treatments.

**Figure 3 plants-12-01773-f003:**
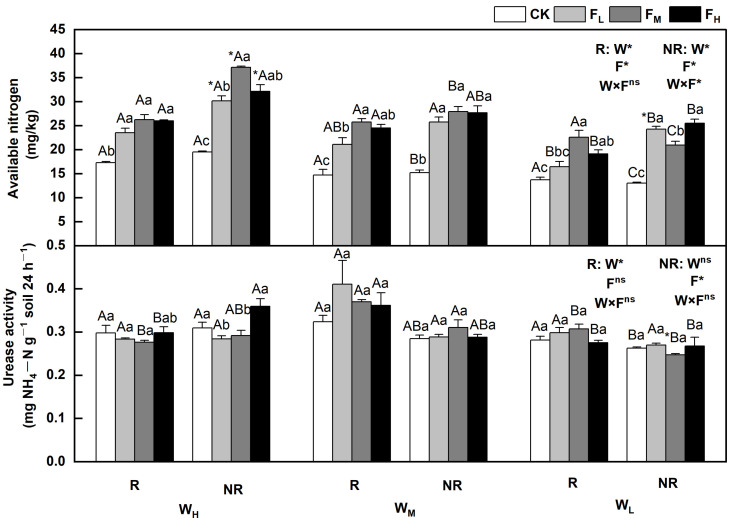
Changes in available nitrogen and urease in the soil under different water and fertilizer treatments. Note: R and NR indicate rhizosphere and bulk soils, respectively. W, F, and W×F indicate the effects of water and fertilizer and their interaction from the results of the ANOVA (*p* < 0.05). Asterisks (*) and ns indicate significant differences. The uppercase and lowercase letters represent the differences between the water and fertilizer treatments. Asterisks before letters indicate significant differences between the rhizosphere and bulk soils.

**Figure 4 plants-12-01773-f004:**
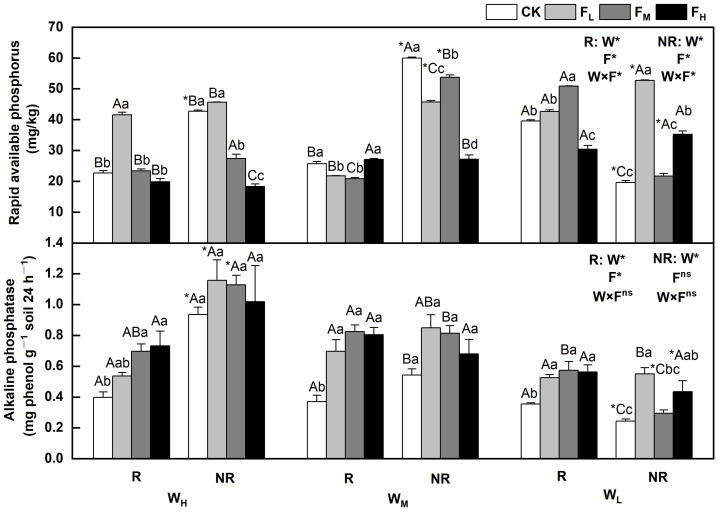
Changes of rapidly available phosphorus and alkaline phosphatase in the soil under different water and fertilizer treatments. Note: R and NR indicate rhizosphere and bulk soils, respectively. W, F, and W×F indicate the effects of water and fertilizer and their interaction from the results of the ANOVA (*p* < 0.05). Asterisks (*) and ns indicate significant differences. The uppercase and lowercase letters represent the differences between the water and fertilizer treatments. Asterisks before letters indicate significant differences between the rhizosphere and bulk soils.

**Figure 5 plants-12-01773-f005:**
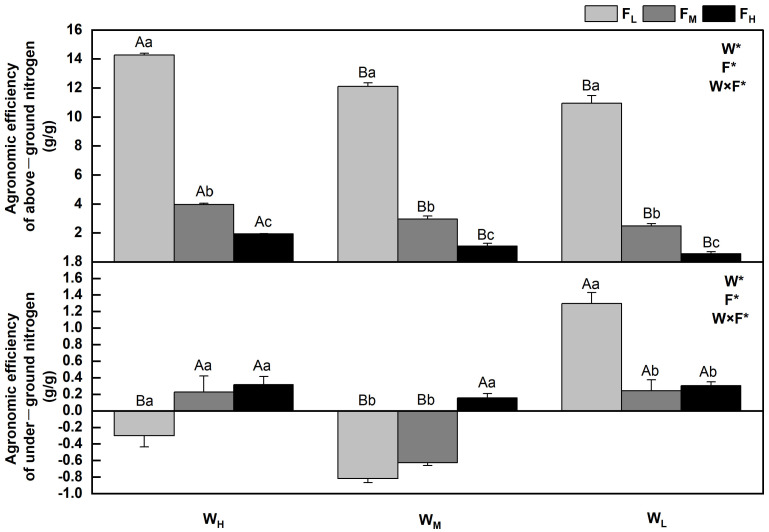
Changes in agronomic efficiency of nitrogen of *Festuca coelestis* under different water and fertilizer treatments. Notes: W, F, and W×F indicate the effects of water and fertilizer and their interaction from the results of the ANOVA (*p* < 0.05). Asterisks (*) and ns indicate significant differences. The uppercase and lowercase letters represent the differences between the water and fertilizer treatments.

**Figure 6 plants-12-01773-f006:**
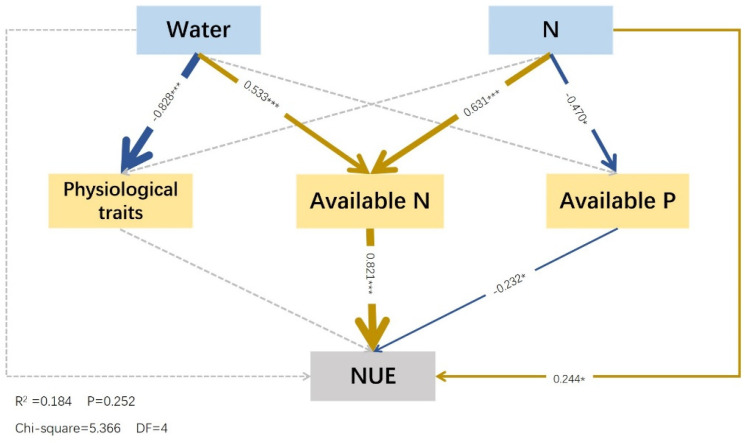
Structural equation model of effects of water and fertilizer on physiological properties and soil nutrients. Notes: The colored and gray arrows represent significant and non–significant relationships, respectively (* *p* < 0.05, and *** *p* < 0.001; only the values of path coefficients with significance are presented in SEM). Positive correlations are indicated by dark blue arrows, and negative correlations are indicated by yellow arrows. The arrow thickness represents the values of path coefficients; the thicker the path, the larger the value. The factors, from top to bottom, were water and fertilizer, physiological characteristics, alkali–hydrolyzed nitrogen, available phosphorus, and nitrogen use efficiency. Physiological traits involve the conversion of multiple factor variables (nonstructural carbohydrates and proline) into a set of variables using PCA.

## Data Availability

Not applicable.

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
