# Peer review of "Festuca coelestis Increases Drought Tolerance and Nitrogen Use via Nutrient Supply–Demand Relationship on the Qinghai-Tibet Plateau"

_plants, 2023, doi:10.3390/plants12091773_

Round 1

Reviewer 1 Report

General comments

All parts of the manuscript have the same problems and strengths. The aim of the project is clear, to help establish the native plants on the Qinghai-Tibet Plateau, in a dry, nutrient deficient environment. Consequently, the effect of water and nutrient supply was investigated.

Although important, the investigation of interactions with other organisms was omitted. Such interactions for example the fungi-plant symbiosis (mycorrhiza), competitions with other plants (weeds). These interactions are decisive in the success of a plant. Even if there were not examined, they should be mentioned.

It should also be emphasized that N treatment was not only N, but NPK treatment.

Similarly, it is not clear (in the whole manuscript) what variables were measured. Four (starch, non-structural carbohydrates, soluble sugars, proline) or three (starch, soluble sugars, proline)? e.g. Fig1. contains four, Material and methods contains three measured variables.

Calculation of nitrogen use efficiency surely clear to the authors but not the readers. The applied model gave interesting results, but requires more description and explanation.

Detailed review

Abstract

Like with general comments.

WhFm abbreviation not explained previously.

“Productivity” was mentioned, but I the text there was no measurements of productivity.

Sentence in lines 27-28 is not clear.

The results of enzyme measurements and the applied model was not mentioned in the abstract.

Introduction

Like with the general comments.

In the second paragraph (lines 50-65) non-structural carbohydrates was not mentioned. But it appeared in line 116. So, its meaning should be defined somewhere. If it was measured and used.

Enzymes (lines 78-100). The description of the urease enzyme is not correct. It hydrolyzes urea. It doesn’t promote the transformation of organic materials (except urea), and doesn’t participate in N2 fixation. For both enzymes, it would be worthwhile to specify how long their lifetime is as immobilized enzyme in the soil.

More reference is needed in the theme of the water-fertilizer coupling.  

And it should also be explained in this chapter why plant biomass was not measured.

Materials and methods

Like with general comments.

This is the part of the manuscript that needs the most clarification. More thorough description of the used soil and the investigation methods is necessary.

4.1. It was mentioned in the abstract and in the text, that the area (Qinghai-Tibet Plateau) is dry and the soils are nutrient deficient. Data of the annual precipitation and a description of soil nutrient supply would be useful.      

4.2.1. The soil type (according to one of the chosen taxonomy), soil pH, soil organic matter (SOM) and carbonate contents are essential for the correct interpretation of the experimental results. The initial values of available nutrient (N, P) concentrations are also important.

With the fertilizer treatments, the description should be consistent, as far as the units are concerned. N, P and K doses can be given in N mg/kg, P2O5 mg/kg and K2O mg/kg units, or in N mg/kg, P mg/kg and K mg/kg units. But not mixed.

4.3.1. A more detailed description is needed, with the proper reference (e.g. Ref 45 is only a book review). The methos of urease measurement is missing.

4.3.2. Method for measurement of non-structural carbohydrates is not mentioned. But Fig1 contains its values. In line 459 “throne” is mistyping, “anthrone” is the proper word.

4.3.3. Root indexes. Detailed description is needed, and it should be highlighted, what were the measured parameters, and what were the calculated ones.

4.4. The characteristic of the used model should be written here.

Results

As with the general comments.

2.1. Non-structural carbohydrates (in lnes 125, 132) is necessary to define what it means, and how it was measured. Fig 1 contains four variables (starch, non-structural carbohydrates, soluble sugars, proline), included non-structural carbohydrates.

2.3. correct subtitle: Effect of water and fertilizer treatments on N and P in the soil

The title of Fig 3 is incorrect, instead of “in Festuca coelestis” the proper one is “in the soil”.

Phrases “urease content” (in Fig 3, and in the text) and “alkaline phosphatase content” (in Fig 4 and in the text) is wrong. Enzyme activity was what was measured. So, urease activity is the correct phrase. And the units of enzyme measurements is also have to change.

2.4. lines 245-254, The description of calculation of agronomic efficiency of N is missing. So, it is difficult to interpret the results obtained.

2.5. in line 268. “in this model”. But we don’t know anything about how the model works.

Discussion

As with general notes.

It’s hard to synthetise the experimental data, because of lot of inaccuracy. I suggest rethinking the discussion and adding a paragraph dealing with the evaluation of initial soil properties at the beginning. The results of fertilization are always multifactorial. The soil properties (and their changes during the experiment) are as important as the demands of the plants.

The changes in the (bulk and rhizosphere) soil and in plant can be evaluated separately, but they are interacting with each other. These interactions can be investigated with the SEM model. But much more data and parameters must be published about how the model works, a graphic model (Fig 6) is not enough.  And the hypothesis that the model justified must also be clearly described.

In summary, I think the manuscript contains many results, but their evaluation nedds to be improved.

Reviewer 2 Report

The introduction is written broadly, but there is a lot of elementary information, more concreteness and raising of the problem is needed.

Sugars, carbohydrates are presented as percentages, from what are the percentages calculated? In some variants, they reach 300%.

At what temperature did the plants grow?

Describe the methods in more detail, how much material you used, how you prepared it for analysis.

I can't find a clear explanation of what CK, WLCK and other abbreviations are.

Conclusions - continuation of the discussion, narrow it down and make it more specific.

Round 2

Reviewer 1 Report

The authors modified the manuscript according to the comments. Disturbing inaccuracies, mainly in the material and method chapter, have been largely corrected. The revised manuscript meets the requirements of Plants.

Best regards

There are a few typos and misspellings in the text (I wrote some of them down), so I recommend looking over the manuscript carefully.

capital letter in line 116

enzyme content – enzyme activity replacements are required in lines between 370-386

word repetitions in line 428

Reviewer 2 Report

After the corrections, the manuscript is of good quality, I have no further comments.

Author Response

Thank you for your guidance and advice to me